# Forecast the Habitat Sustainability of *Schoenus ferrugineus* L. (Cyperaceae) in the Southern Urals under Climate Change

**DOI:** 10.3390/plants13111563

**Published:** 2024-06-05

**Authors:** Nikolay Fedorov, Albert Muldashev, Oksana Mikhaylenko, Svetlana Zhigunova, Elvira Baisheva, Pavel Shirokikh, Ilnur Bikbaev, Vasiliy Martynenko

**Affiliations:** 1Ufa Institute of Biology, Ufa Federal Research Centre, Russian Academy of Sciences, Ufa 450054, Russia; muldashev_ural@mail.ru (A.M.); zigusvet@yandex.ru (S.Z.); elvbai@mail.ru (E.B.); shirpa@mail.ru (P.S.); ilnur.bikbaev.90@mail.ru (I.B.); vb-mart@mail.ru (V.M.); 2Department of General and Analytical Chemistry, Ufa State Petroleum Technological University, Ufa 450064, Russia; trioksan@mail.ru

**Keywords:** *Schoenus ferrugineus*, relict species, MaxEnt, climate change, calcareous mires, habitat suitability

## Abstract

An analysis of the current potential range of the Pleistocene relict plant species *Schoenus ferrugineus* and modeling of changes in its future range under moderate (RCP4.5) and strong (RCP8.5) climate change in the middle and second half of the 21st century were carried out. The MaxEnt program was used for modeling. Climate variables from CHELSA Bioclim, the global digital soil mapping system SoilGrids, and a digital elevation model were used as predictors. Modeling has shown that climate change will lead to a significant reduction in the suitability of *S. ferrugineus* habitat conditions by the mid-21st century. The predicted changes in the distribution of habitats of *S. ferrugineus*, a diagnostic species of calcareous mires and an indicator of their ecological state, indicate a possible strong transformation of wetland complexes in the Southern Urals region even under moderate climate change. A reduction in the distribution of *S. ferrugineus* at the eastern limit of its range will also be facilitated by more frequent extreme droughts. To maintain the distribution of *S. ferrugineus* on the eastern border of its range, a number of measures are proposed to mitigate the negative consequences of climate change, contributing to the preservation of the hydrological regime of calcareous mires.

## 1. Introduction

*Schoenus ferrugineus* L. (Cyperaceae Juss.) is a perennial plant species that grows mainly on the calcareous fens [1,2,3,4]. Calcareous mires with this species are usually located on gentle terraces in river valleys and lake basins. These mires are characterized by a combination of tufa (travertine) deposition at the surface of the mire and peat formation within it [5]. Travertine deposition prevents acidification of the mire and sustains populations of basiphilous species [6]. In these mires, the plants are deficient in plant-available phosphorus due to its immobilization by iron and calcium complexes. Phosphorus availability appears to limit the productivity of plant communities in calcareous mires [7], but in managed (mown) mires, the productivity appears to be co-limited by phosphorous and nitrogen availability [8].

*Schoenus ferrugineus* shoots have a two-year growth cycle. New shoots appear in autumn and reach a height of 3–7 cm [9,10]. This development represents an adaptation strategy for infertile habitats because it allows nutrients (especially nitrogen and phosphorus) to move from old shoots to new ones [9]. *S. ferrugineus* is reproduced mainly by seeds. The seed productivity of this species is low and usually does not exceed six seeds per inflorescence [11]. Flowering occurs in mid-July, even in the northernmost populations. The rate of fruiting is very low (10–32%), and the remaining seeds do not reach maturity [12,13]. Most seeds are dormant in spikelets for over a year, and 20% of seeds may germinate even in the third year [13]. However, germination in pot culture may reach 50%, with a 10–20% delay of up to two years [13]. The population size of this species can vary greatly depending on weather conditions [3,13]. Population stability is achieved through uneven seed germination rates [13], some of which germinate the following year, which ensures the stabilization of the population in favorable years. The long-distance dispersal of seeds is limited because most of them remain in the inflorescence for one year after flowering [10]. Preserved seeds germinate when they come into contact with a moist soil surface [13].

In Central Europe, *S. ferrugineus* and the plant communities dominated by this species are widespread in the foothills of the Alps in Germany, Switzerland, and Italy [14,15,16]. This species is common in Scandinavia and is also found in Eastern Europe [17,18,19]. In Europe, the northern border of its range is situated approximately 60 km north of the Arctic Circle [3]. The eastern border of the main range of calcareous fens with this species is in Finland, the Baltic States (Latvia, Lithuania, Estonia, Poland), Slovakia, and Ukraine [20,21].

Outside the main range, several small populations were found in Belarus [22], northwestern Russia (the Republic of Karelia, Murmansk, Leningrad, Pskov, Novgorod, Vologda and Arkhangelsk regions) [3,23,24,25], in the central part of European Russia (Bryansk, Tula, Moscow and Samara regions, the Republic of Tatarstan), and the Southern Urals (the Republic of Bashkortostan and Chelyabinsk region) [26,27,28,29,30]. Further eastwards in the river valleys of Western Siberia, moderately calcium-rich sedge-moss fens have been found, but *S. ferrugineus* is absent there [31]. Thus, in the Southern Urals, this species is located at the eastern limit of its range [4,32,33]. Some habitat types of base-rich (calcareous) fens were categorized as endangered, falling among 10% of the most threatened European habitats [34]. The rarity of calcareous fens, as well as their disturbances due to drainage, has led to the fact that *S. ferrugineus* becoming rare and is listed in the Red Data Books of the Russian Federation [35], as well as some European countries, e.g., Finland, Norway, Ukraine, etc. [36,37,38,39,40].

Calcareous fens formed along the perimeter of melting glaciers in the Late Pleistocene or Early Holocene, making them refugia for a number of relict plant species, including *S. ferrugineus* [12,41,42,43,44]. However, *S. ferrugineus* is not included in the IUCN Red List of threatened species [45]. Calcareous fens are actively fed by carbonate groundwater [46,47] and are among the most vulnerable ecosystems to climate change [48,49,50]. With increasing summer drought conditions, groundwater levels may decline, and the wetland complexes will degrade with the establishment of different plant communities.

*S. ferrugineus* is one of the diagnostic species of calcareous fens, which are listed in the network of protected areas Natura 2000 covering Europe’s most valuable and threatened species and habitats (EU-FFH 7230) [51,52]. In Europe, *S. ferrugineus* is largely restricted to calcareous areas [53] and typically grows in soligenous (spring) fens in alpine and arctic regions [54]. However, despite its apparent predilection for calcareous mire conditions, *S. ferrugineus* is not exclusively associated with calcic habitats and can grow in less base-rich environments. This is particularly noticeable in some Baltic mires, where it may occur together with *Andromeda polifolia* L., *Carex lasiocarpa* Ehrh., *C. limosa* L., *Myrica gale* L., *Rhynchospora alba* (L.) Vahl, *Salix rosmarinifolia* L., *Trichophorum cespitosum* (L.) Hartm. and *Vaccinium oxycoccos* L. [53,55]. It can also be found in open stony meadows and even in crevices among wet rocks [56]. *S. ferrugineus* is most often found in the plant communities belonging to the alliance *Caricion davallianae* Klika 1934 of the order *Caricetalia davallianae* Br.-Bl. 1949 (the class *Scheuchzerio-Caricetea fuscae* Tx. 1937) [57]. The communities of this alliance include mineral-rich fen vegetation on both calcareous tufa-forming springs and peat substrates developed on limestone, calcareous sedimentary or metamorphic rocks, and ultrabasic crystalline rocks. The herb layer of these communities consists mainly of calciphilous graminoids (e.g., *Carex davalliana* Sm., *C. hostiana* DC., *Eleocharis quinqueflora* (Hartmann) O. Schwarz, *Eriophorum latifolium* Hoppe, *S. ferrugineus*) and herbs such as *Parnassia palustris* L., *Pinguicula vulgaris* L., *Primula farinosa* L. and *Tofieldia calyculata* (L.) Wahlenb. The moss layer consists of *Campylium stellatum* (Hedw.) C.E.O.Jensen, *Drepanocladus revolvens* (Sw. ex anon.) Warnst., *Palustriella commutata* (Hedw.) Ochyra, *Philonotis calcarea* (Bruch et al.) Schimp., etc. [57].

In the Murmansk region (Russian Federation) and the Republic of Karelia, *S. ferrugi-neus* was found in the communities of the association *Trichophoro-Schoenetum ferruginei* Görs 1964 [58,59,60]. In the Republic of Bashkortostan (the Southern Urals), calcareous fens with *S. ferrugineus* are very rare and belong to the associations *Primulo–Schoenetum ferruginei* (Koch 1926) Oberdorfer 1957 and *Caricetum paniceo–lepidocarpae* Braun 1968 [20,26]. The floristic composition of these communities is similar to that at the northern limit of the *S. ferrugineus* distribution area, which is caused by carbonate-rich groundwater [26].

Analysis of the composition of peat deposits within these communities has shown that their floristic composition has not practically changed from the start of peat formation to the present time. Only small changes in the proportion of species in the composition of palaeo-communities have been observed [61]. Calcareous fens have been recognized as local hotspots of biological diversity, but little is yet known about the effects of projected climate change on the plant communities of these ecosystems [62]. Since the early 20th century, these mires have been heavily influenced by anthropogenic activities—drainage for peat extraction and use as agricultural land [63]. Even in the absence of drainage, lowering water tables may increase the spread of *Phragmites australis*, negatively affecting the biodiversity of light-loving wetland species. Grazing suppresses *Phragmites australis* and can have a positive impact on relict species [64]. It is thought that being Pleistocene relicts, communities with *S. ferrugineus* are likely to be threatened by winter warming [61]. This is especially true for species at the edge of their range [65]. The possible deterioration of *S. ferrugineus* habitat conditions under climatic changes can, therefore, be considered unfavorable changes in the hydrological regime of unique carbonate mires.

According to different climate change scenarios, the current potential and future distribution can be determined through machine learning methods by means of the layers created using the records of point areas where species are currently located and digital bioclimatic data regarding these areas. There are various species distribution models, such as CLIMEX, GARP, and MaxEnt, that estimate the impact of environmental and climate change on species and ecosystems. Among these modeling approaches, MaxEnt has advantages because it uses both presence data and categorical data as input. In addition, it tests forecast precision, it is always stable and reliable, even in small sample sizes and with gaps, and it directly produces a spatially open habitat suitability map and evaluates the significance level of individual environmental variables using a built-in jackknife test [66,67]. The aim of this work was to model the potential range of *S. ferrugineus* and changes in its habitat conditions in calcareous mires located at the eastern limit of the species’ range under moderate and strong climate change scenarios using the maximum entropy method.

## 2. Materials and Methods

To model the potential range of *S. ferrugineus* in the Southern Urals, we used 51 known geo-referenced locations in Russia based on herbarium collections (UFA) and literature data [26,27,68], 34 of which were located on the eastern border of the range. In addition, to better cover the environmental features of *S. ferrugineus* habitats, locations in Western, Central, and Northern Europe were used. These locations were at least 10 km apart and were randomly selected from the GBIF [69]. A total of 203 locations were sampled within the core range using the spThin module in R [70]. Additionally, data from the Regional Red Data Books were also used to interpret the results. The habitat locations of *S. ferrugineus* used in the modeling are shown in Figure 1.

Maximum entropy modeling software (MaxEnt v3.4.4) was used to assess the changes in *S. ferrugineus* habitat conditions under climatic changes [71]. The BIOCLIM CHELSA (Climatologies at high resolution for the earth’s land surface areas) set was used in the modeling as climatic variables [72,73]. In addition to climatic variables, characteristics of the digital elevation model GMTED2010 were used in modeling [74]. In addition, Soil Organic Carbon (SOC) layers in the 5–15 cm soil horizon and Organic Carbon Stocks in the 0–30 cm soil horizon (OCS) taken from SoilGrids 2.0, which recognizes the species’ relationship with the environment were used for ecological calibration [75]. The raster layers of the environmental data were restricted to Eurasia. We generated a Pearson correlation matrix of environmental predictors. In the case of a correlation coefficient greater than or equal to 0.7, one of the variables was excluded to prevent multicollinearity and model overfitting [76]. At the same time, preference for further use was given to variables with a greater contribution to the model identified at the preliminary stage and, other things being equal, to parameters reflecting quarterly rather than monthly characteristics of temperature and precipitation. We used the following MaxEnt settings: maximum iterations—1000, replicates—5, and output format—“cloglog”. The “cloglog” output gives a score between 0 (completely unsuitable) and 1 (completely consistent with the species or community optimum) for habitat suitability.

The RCP4.5 and RCP8.5 scenarios corresponding to moderate and strong atmospheric greenhouse gas concentrations for two time periods, 2041–2060 (2050) and 2061–2080 (2070), respectively, were used to project changes in *S. ferrugineus* habitat suitability under climate change. The future potential range was modeled using an ensemble of four climate change models: CCSM, NorESM1-M, MIROC-ESM, and INMCM4 [77,78,79,80,81]. The selection of climate change models was carried out in accordance with the recommendations of C.F. McSweeney and B.M. Sanderson [82,83]. To assess the resistance of *S. ferrugineus* habitats to extreme droughts, the frequency of which may increase under climatic changes, we analyzed the change in the NDVI in from July to early August in the drought year of 2010 in comparison with that in 2009 [84].

The AUC indicator was used for the statistical evaluation of the model [85]. We applied the «10 percentile training presence» threshold as the lowest limit for habitat suitability [86,87]. In the final models, the habitat suitability with values above the lowest limit for habitat suitability was divided into three equal groups: low, medium, and high.

## 3. Results

### 3.1. The Current Potential Range of Schoenus ferrugineus

The resulting model of the current potential range of S. ferrugineus on the eastern border of the species range has an AUC of 0.97, which corresponds to the high quality of the model [85]. Table 1 shows the contribution of the variables to the model of the current potential range of S. ferrugineus. The following four environmental variables had the greatest contributions: precipitation amount of the driest month (Bio14), mean daily minimum air temperature of the coldest month (Bio6), soil organic carbon content (SOC), and mean daily air temperature of the warmest quarter (Bio10).

Figure 2 shows the model of the modern potential range of *S. ferrugineus* on the eastern border of its distribution area and adjacent territories. The potential range map shows habitats of low suitability (0.36–0.57), medium suitability (0.58–0.79), and high suitability (0.80–1.00). Calcareous fens are unevenly distributed from the Volga Upland to the Southern Urals.

In the Southern Urals, highly and moderately suitable habitat conditions for *S. ferrugineus* are confined to the Mesyagutovskaya forest steppe area in the Republic of Bashkortostan and the adjacent territory of the Sverdlovsk Region (Figure 2 and Figure 3). The same conditions were also revealed in the Chelyabinsk Region, i.e., the zone of contact between the forest steppe of the northeastern foothills of the Southern Urals and mountain forests bordering the Republic of Bashkortostan.

In the Cis-Urals, the locations of the *S. ferrugineus* habitats with high and moderate suitability are confined to the forest steppe of the Bugulma-Belebey Upland located in the parts of the Republic of Bashkortostan, the Republic of Tatarstan, the Samara and Orenburg Regions. In addition, *S. ferrugineus* is sporadically distributed in the broad-leaved forests of the western slope of the Southern Urals. A small number of habitats with moderately suitable conditions is located in the northeastern foothills of the Southern Urals in the contact zone between the forest steppe and mountain forests, namely in the depression between the Uraltau and Irendyk ridges. The habitats with low suitability conditions are located on the southern spurs of the Southern Urals, as well as in the northern and southern parts of the Bugulma-Belebey Upland.

In the neighboring territories of the Volga Upland, Penza, Ulyanovsk, and Samara Regions, where *S. ferrugineus* is listed in the regional Red Data Books [88,89,90], there are predominantly low habitat conditions, which are confined mainly to the forest steppe zone.

It should be noted that the potential range of *S. ferrugineus* distribution is wider than the current range at the eastern border. According to the model, low habitat conditions are observed in the territories of the Nizhny Novgorod, Kirov, and Sverdlovsk Regions, in the Republics of Mordovia, Mari-El, and Chuvash, as well as in the Perm Krai. *S. ferrugineus* was not found in these areas. Still, other species typical of calcareous mires (*Carex panicea*, *Dactylorhiza incarnata*, *Eriophorum latifolium*, etc.) are listed in the Red Data Books of these regions [91,92,93,94,95,96,97].

### 3.2. Suitability of S. ferrugineus Habitat Conditions in Wetland Complexes at the Eastern Limit of Current Species Distribution

At the eastern limit of *S. ferrugineus* range, 44 calcareous mires have been identified, of which *S. ferrugineus* occurs in only 30 localities. Figure 4 shows the results of habitat suitability modeling in calcareous mires with and without the presence of *S. ferrugineus*. Highly suitable habitat conditions for this species were identified only in two wetland complexes in the Mesyagutovskaya forest steppe. Moderately suitable habitat conditions were found in 12 mires in the Mesyagutovskaya forest steppe and in four fens in the area of mountain light coniferous forests on the northeastern slope of the Southern Urals. The only locality with moderately suitable habitat conditions (Uskyurt Mire) is located on the Bugulma-Belebey Upland in the Bashkir Cis-Urals. Low habitat conditions are observed in 11 mires, most of which are located in the Bashkir Cis-Urals. According to the modeling, unsuitable habitat conditions in which *S. ferrugineus* nevertheless was found were observed in two mires in the Bashkir Cis-Urals and in one fen in the western part of the Bugulma-Belebey Upland. In addition, unsuitable habitat conditions are noted in one site located on the Volga Upland, which is 400 km away from the mires of the Bugulma-Belebey Upland (Figure 3).

In the Bashkir Cis-Urals, 13 of 14 calcareous mires with the absence of *S. ferrugineus* are located (Figure 4), of which only three mires have moderately suitable habitat conditions for this species. These mires are small (with areas of 2.5 to 5 hectares) and are located on the Bugulma-Belebey Upland on gentle southern and southwestern slopes.

### 3.3. Changes of Habitat Suitability of S. ferrugineus in Wetland Complexes at the Eastern Border of Species Range under Future Climate Change

Under moderate climate change (RCP4.5), by the middle of the 21st century, highly suitable habitats of *S. ferrugineus* will disappear at the eastern border of the species range (Figure 4). No great changes in the suitability of *S. ferrugineus* habitat conditions are predicted in the mires of the Mesyagutovskaya forest steppe and in the northeastern slope of the Southern Urals. Moderately suitable habitat conditions will be maintained in 11 mires in the Mesyagutovskaya forest steppe and in four mires in the zone of mountain light coniferous forests of the eastern slope of the Southern Urals. In the rest of the study area, a sharp decline in habitat suitability was predicted (Figure 5a). On the Bugulma-Belebey Upland, low-suitability habitat conditions will persist in only two mires (Figure 5a). Under moderate climate change (RCP4.5), by the middle of the 21st century, in calcareous mires where *S. ferrugineus* is not found, low habitat suitability is predicted in four localities of the Bugulma-Belebey Upland and in one locality in the Mesyagutovskaya forest steppe (Figure 5a).

In the second half of the 21st century, the habitats with moderate suitability will be preserved only in four localities in the Mesyagutovskaya forest steppe and in one locality in the zone of the mountain light coniferous forests of the eastern slope of the Southern Urals. On all other calcareous mires, habitat conditions will decline to low suitability. On the Bugulma-Belebey Upland, only two mires will retain low habitat conditions (Figure 5b).

Under a strong climate change scenario (RCP 8.5) in the mid-21st century, in the mires with *S. ferrugineus* populations, changes in habitat suitability are projected to be similar to those under a moderate climate change scenario in the second half of the 21st century. Moderately suitable habitats will completely disappear by the second half of the 21st century. Low suitable habitat conditions will remain only in three mires in the Mesyagutovskaya forest steppe and one mire in the zone of the mountain light coniferous forests of the eastern slope of the Southern Urals.

Under the scenario of strong climate change (RCP 8.5), in calcareous mires where *S. ferrugineus* was not found, only two mires of the Bugulma-Belebey Upland are predicted to retain low habitat conditions by the middle of the 21st century. By the second half of the 21st century, suitable habitat conditions will disappear in all these mires.

## 4. Discussion

### 4.1. Current Potential Range of Schoenus ferrugineus

Currently, the main range of *S. ferrugineus* and the plant communities with this species are located in Central Europe [14] (Figure 3). Outside this area, small populations of *S. ferrugineus* are found in the European part of Russia, reaching the Mesyagutovskaya forest steppe and the northeastern foothills of the Southern Urals (Figure 3) [3,22,23,24,25,26,27,28,29,30]. Several reasons for the fragmentation of the *S. ferrugineus* range can be identified. First, calcareous mires are a very rare and vulnerable habitat type confined mainly to carbonate areas [53]. In Russia, calcareous mires do not have a continuous distribution throughout the country.

Secondly, the fragmentation of the *S. ferrugineus* range was caused by climate change during the Holocene, when the increase in temperature and decrease in precipitation resulted in the expansion of steppes and forest steppes in European Russia [98].

The third and, perhaps, the main reason for the reduction and fragmentation of the *S. ferrugineus* range is human impact, i.e., the drainage of mires and deforestation in the catchment areas, which affects the water supply of wetland complexes. The maintenance of a constant high level of groundwater is the most important condition for the preservation of calcareous mires. Even moderate drainage leads to a significant increase in the biomass of mesophytic herbs and a decrease or even disappearance of typical basiphilous mire species [7]. At the eastern border of the *S. ferrugineus* range, the distribution of this species is limited by the rarity of calcareous mires, a significant part of which is degraded as a result of human impact. The same reason for the decrease in the area of spring mires is also indicated for the mountainous regions of the Czech Republic [99,100].

According to the modeling, the habitats with low suitability were identified in most of the territories of the Republic of Tatarstan, Ulyanovsk, and Samara regions of Russia. However, *S. ferrugineus* was not found there. This is likely due to the extensive drainage of mires in this area during the 20th century [101].

The fourth reason is the overgrowth of mires with woody vegetation due to the natural dynamics of wetland complexes in the forest zone. For instance, the calcareous mire Lagernoe in the Republic of Bashkortostan is currently overgrown with pine (*Pinus sylvestris* L.), and the population of *S. ferrugineus* survives only in the open central part of this mire. In addition, the lowering of groundwater level may increase the abundance of *Phragmites australis* (Cav.) Trin. ex Steud., which negatively affects the biodiversity of light-demanding mire species such as *S. ferrugineus* [64]. The low seed productivity of *S. ferrugineus* [3,10,13] may cause other species with a wider ecological range and competitive advantages to replace it under unfavorable conditions [102].

A study of environmental factors influencing the distribution of *S. ferrugineus* in calcareous mires of the Southern Urals region revealed a significant influence of altitude and a small influence on summer precipitation [103]. However, the results of our modeling showed that three climate variables had the greatest contribution to the model: Bio14—precipitation amount of the driest month, Bio6—mean daily minimum air temperature of the coldest month, and Bio10—mean daily air temperatures of the warmest quarter. In the study area, the driest months are winter ones. Snow accumulation in winter affects the groundwater level and water saturation of streams feeding wetland complexes. Thus, the winter precipitation has a positive effect on the habitat suitability of *S. ferrugineus*. The contribution of winter temperatures to the model of the potential range of *S. ferrugineus* is not fully understood. However, it can be assumed that the ecological optimum of this species on the eastern border of its range is associated with colder habitat conditions, which are found in the Mesyagutovskaya forest steppe and on the eastern slope of the Southern Urals. This is consistent with the literature data, which show that this species prefers mountainous and cold habitats [54]. The contribution of the summer month temperature to the model can be related to evapotranspiration, which affects the water supply of the mires, as well as the pH and carbonate concentration of the mire water. A significant contribution to the model of Soil Organic Carbon content (SOC 5-15) is likely due to the species’ preference for carbon-rich peat soils.

### 4.2. Changes in Habitat Suitability of S. ferrugineus in Wetland Complexes at the Eastern Border of Species Range Underclimate Change

With climate change, much of the study area is expected to experience lower summer precipitation and higher temperatures, which will inevitably lead to lower wetland water levels [104,105,106]. Under moderate climate change (RCP4.5), modeling results do not predict a strong decrease in habitat suitability for *S. ferrugineus* in the Mesyagutovskaya forest steppe and in the northeastern part of the Southern Urals by the middle and to the end of the 21st century. In the other parts of the study area (Bugulma-Belebey Upland, the Bashkir Cis-Urals), a sharp decline in habitat suitability is predicted as early as in the first half of the 21st century. This finding is consistent with literature data on the strongest changes in habitat conditions of rare species in the first half of the 21st century [65,107]. Under the RCP4.5 scenario, in the second half of the 21st century, suitable habitat conditions for *S. ferrugineus* will remain only in some calcareous mires of the Mesyagutovskaya forest steppe and on the eastern slope of the Southern Urals. According to the strong climate change scenario (RCP8.5), already in the middle of the 21st century, habitat suitability for this species will be strongly reduced in the Mesyagutovskaya forest steppe and will completely disappear in the Southern Urals but will persist in two habitats where the species is not currently found. In the second half of the 21st century, only habitats with low suitability will persist in the Mesyagutovskaya forest steppe and on the eastern slope of the Southern Urals.

Our results are consistent with the literature data on the impact of climate change on calcareous mires in Spain and Slovakia [108], in which it was shown that the main threat to wetland conservation is changes in precipitation rather than temperature. It is also noted that decreasing precipitation is the greatest threat to the existence of spring-fed wetland ecosystems because they depend on groundwater supplies [43,109,110]. Currently, severe reduction and substantial transformation of calcareous mires have been reported in the UK [111], in the German and Dutch lowlands [50,112,113], in the Swiss Alps foothills [114,115], in the Czech Republic [116] and even in Scandinavia [117,118]. However, modeling the effects of climate change on carbonate mires in Austria and the Western Carpathians revealed that they would lose small areas [47,50].

MaxEnt modeling accounts for climate change but does not account for the increased frequency of extreme events, such as drought, which may contribute significantly to the decline of the *S. ferrugineus* distribution. The spring-summer drought of 2010 was the worst in the Southern Urals in recent decades. Figure 6 shows that in July 2010, during the drought period, the NDVI of the herb layer in most wetland communities of the Bugulma-Belebey Upland decreased to the level of 0.29, which corresponds to almost dried vegetation. This makes it possible to explain the absence of *S. ferrugineus* in the calcareous mires with medium and low habitat suitability in the southeastern part of the Bugulma-Belebey Upland by the high frequency of droughts (Figure 5). In addition, seed survival in the soil is thought to be improved under moist conditions [119,120], while frequent drying of the upper soil layer increases germination and reduces the seed bank [121]. Consequently, recurrent droughts will deplete the seed bank of *S. ferrugineus*. Thus, the increasing frequency of droughts will greatly affect the future distribution of this species.

Thus, the predicted changes in the distribution of habitats of *S. ferrugineus,* which is a diagnostic species of calcareous mires, may be used as an indicator of the ecological condition of these ecosystems, pointing to a possible strong transformation of the wetland vegetation in the Southern Urals region even under moderate climate change in the future. Projected future higher temperatures and lower summer precipitation could threaten these ecosystems by reducing the active flow of groundwater enriched with dissolved carbon dioxide [122]. The development of effective management strategies to protect or restore these widely threatened habitats requires knowledge not only of recent ecological functioning but also of their long-term history [105,123].

## 5. Conclusions

The spontaneous occurrence of new *S. ferrugineus* localities in secondary wetlands was recorded in Ukraine [124,125]. At the same time, populations of this species were restored at the expense of nearby mires. However, such processes cannot occur in modern agricultural or industrial landscapes, where calcareous mires have a scattered island-like distribution, making them prone to extinction [126]. Our study showed that *S. ferrugineus* habitats are unstable under climate change. Despite some limitations of the method, which does not take into account the frequency of extreme droughts, the modeling showed a decrease in the number of mires with suitable habitat conditions for the *S. ferrugineus*. Therefore, in order to conserve this species at the eastern border of its range, it is necessary to develop a strategy to mitigate the negative effects of climate change, including the following measures:Preservation of the hydrological regime of landscapes and vegetation in areas adjacent to calcareous mires.Preventing long-term flooding of *S. ferrugineus* habitats. In the Republic of Bashkortostan, an example of secondary flooding under climate change is the Arkaulovskoye mire, where the populations of 25 rare plant species, including *S. ferrugineus,* grow [104]. However, upon rewetting with alkaline-rich groundwater, control is necessary to avoid excessive groundwater levels where *Typha latifolia* overgrowth is possible, which may lead to *S. ferrugineus* extinction [127]. At the same time, the prevalence of *Phragmites australis*, which also displaces *S. ferrugineus*, may increase at low water tables [7].In wetland restoration, the removal of topsoil to restore low-nutrient conditions and the introduction of target species, such as by hay transplanting, can help restore *S. ferrugineus* populations [122,128,129]. Without continuous removal of phytomass, only limited success in restoring the characteristic species composition is possible [130]. In this respect, haymaking is the preferred method because it ensures regular removal of nutrients. Subsequent low phytomass production is an important prerequisite for the survival of light-loving, low-growing wetland species [131,132,133].

The integrated use of these measures will contribute to the conservation of *S. ferrugineus* habitats. The promise of this research may be to establish a system for monitoring calcareous mires and changes in their floristic composition in response to future climate change, as well as to develop measures for the conservation of *S. ferrugineus* at the eastern limit of its range. Particular attention should be given to the study of groundwater level fluctuations in extreme dry years.

## Figures and Tables

**Figure 1 plants-13-01563-f001:**
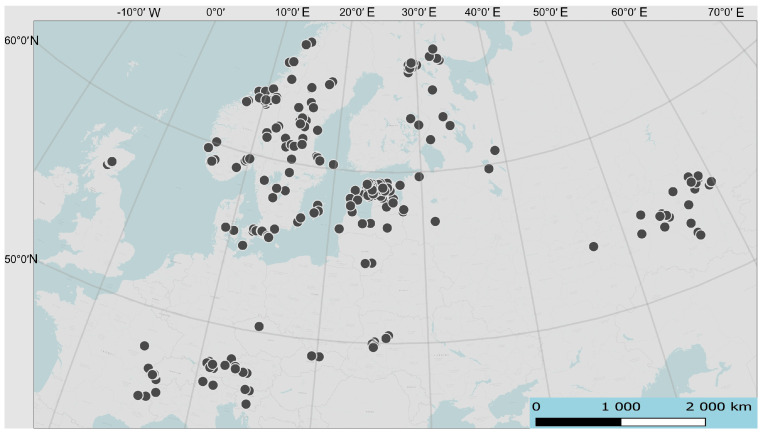
Localization of Schaenus ferrugineus geo-referenced locations selected for modeling.

**Figure 2 plants-13-01563-f002:**
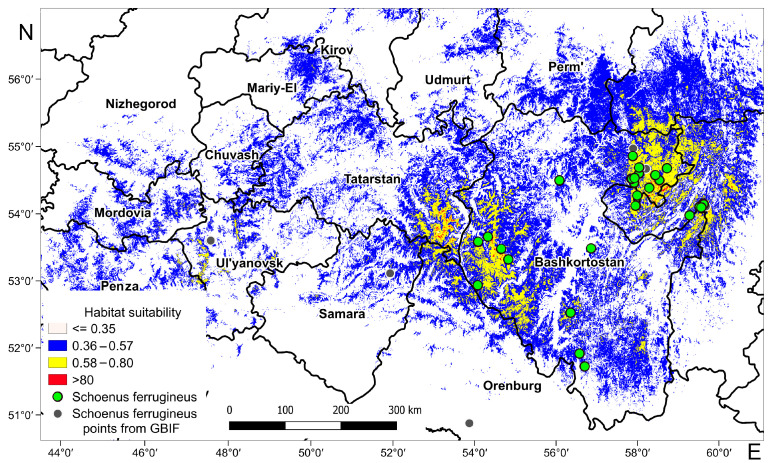
Current potential range of *Schoenus ferrugineus* at the eastern limit of its distribution. Suitability of habitat conditions in fens: blue—low suitability (0.36–0.57), yellow—medium suitability (0.58–0.79), and red—high suitability (0.80–1.00).

**Figure 3 plants-13-01563-f003:**
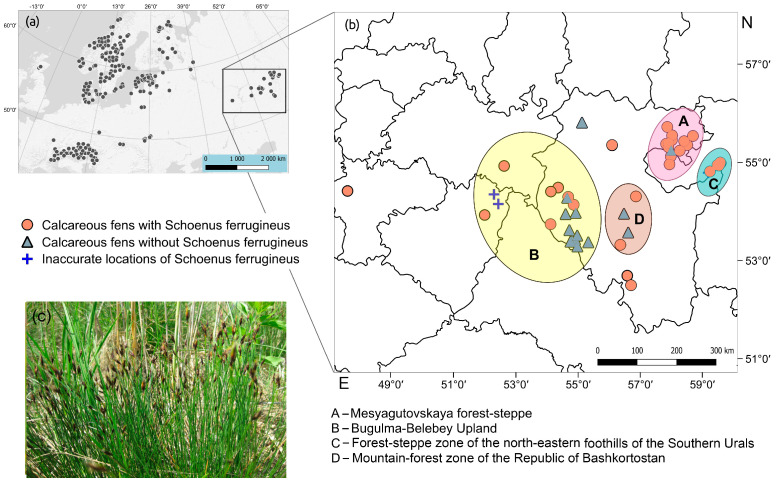
Distribution of calcareous fens in the European and on the eastern border of the range of *Schoenus ferrugineus:* (**a**)—locations of *Schoenus ferrugineus* in Europe; (**b**)—locations of *Schoenus ferrugineus* at the eastern border of its range; (**c**)—photo of *Schoenus ferrugineus*.

**Figure 4 plants-13-01563-f004:**
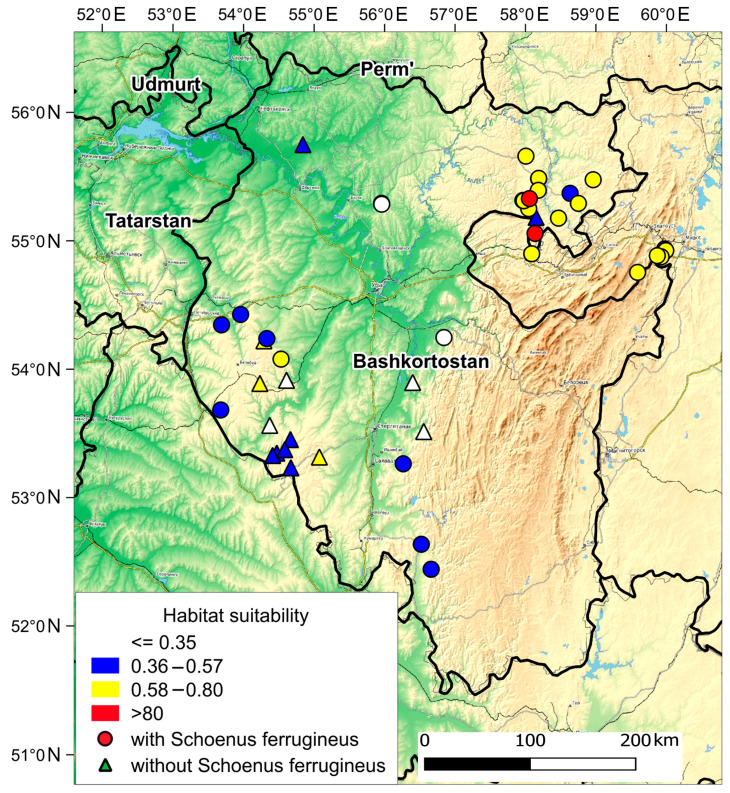
Suitability of *Schoenus ferrugineus* habitat conditions on the calcareous mires in the Southern Urals.

**Figure 5 plants-13-01563-f005:**
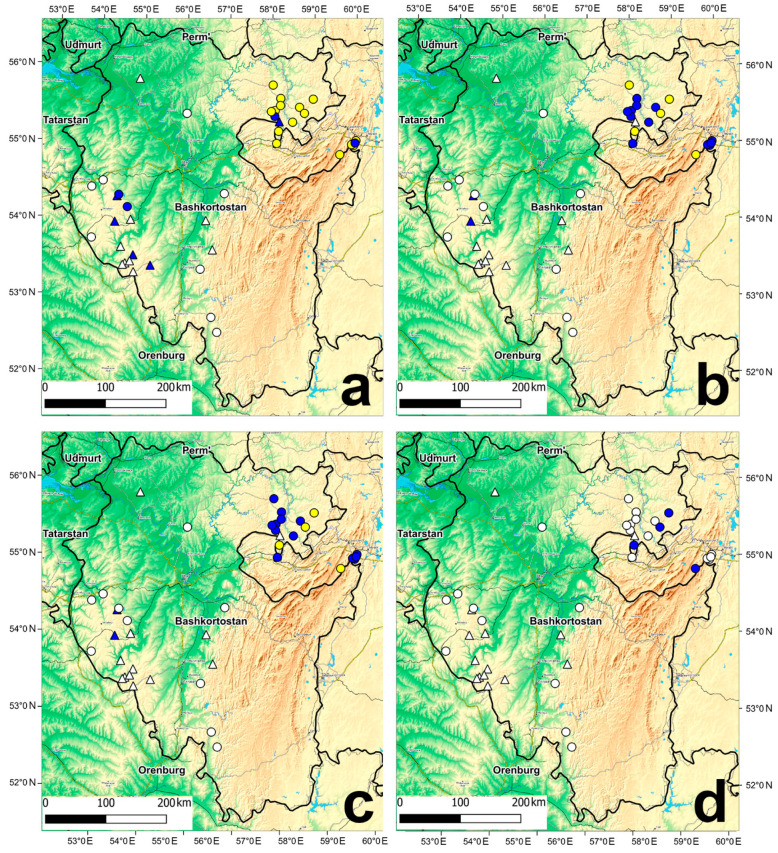
Predictions of the habitat suitability of the locations of *Schoenus ferrugineus* in the calcareous mires by 2050 and 2070 under the moderate (RCP4.5) (**a**,**b**) and the strong (RCP8.5) (**c**,**d**) climate change scenarios. Notes: Habitat conditions: blue circles—with low suitability (0.36–0.57), yellow circles—with medium suitability (0.58–0.79), and wite circles—unsuitable habitats (less than 0.36).

**Figure 6 plants-13-01563-f006:**
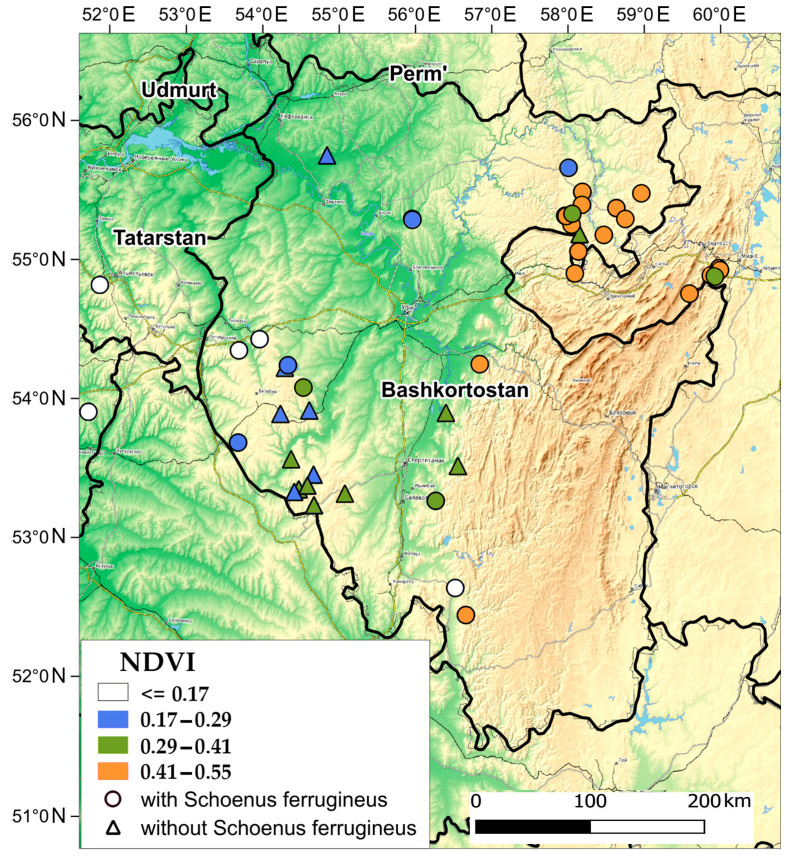
NDVI of the herb layer of plant communities of calcareous mires with the presence and absence of *Schoenus ferrugineus* during the spring–summer drought of 2010.

**Table 1 plants-13-01563-t001:** Contribution of environmental variables to the model of the potential range *Schoenus ferrugineus*.

Code	Environmental Variables	Percent Contribution	Permutation Importance
Bio14	Precipitation amount of the driest month	49.4	7.7
Bio6	Mean daily minimum air temperature of the coldest month	22.4	51.3
SOC	Soil organic carbon content in the fine earth fraction in the 5–15 cm soil layer	12.3	10.1
Bio10	Mean daily air temperatures of the warmest quarter	6.3	18.9
Bio2	Annual average daily temperature amplitude	3.7	3.9
hmax	Maximum elevation above sea level	2.5	2
hmax-min	Difference between maximum and minimum elevation, m	2.3	3.5
OCS	Organic carbon stocks	0.7	0.4
Bio15	Seasonality of precipitation	0.4	0.2
Bio3	Isothermality	0.2	1.6

## Data Availability

The data presented in this study are available on request from the corresponding author.

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
