# Peer review of "Forecast the Habitat Sustainability of Schoenus ferrugineus L. (Cyperaceae) in the Southern Urals under Climate Change"

_plants, 2024, doi:10.3390/plants13111563_

Round 1
Reviewer 1 Report
Comments and Suggestions for Authors
The basis for the study needs to be much more clearly explained in the introduction. The importance of the study is suggested, but there is no background on the general investigative approach to be used. This would be aided if paragraph topics were clearer and opening sentences focused on the general topic of the paragraph.
Insufficient explanation is given to make the methods section interpretable. For example: MaxEntv3.4.4 was used to assess the change of habitat conditions under climatic changes (Line 374). How was it used? What were the input variables and data sources? Explain the rationale for using it. Making the structure of the approach clearer would then make the results easier to follow. It would then be easier to see why the conclusions are justified.
Other comments from the introduction:
Line 32 Rather than “They are characterized…” be specific about what is being referred to, i.e. These These calcareous mires are characterized…
Line 46. What is the fruiting rate a proportion of? Do 10 – 32% of flowers form a fruit? If 20% of seeds germinated in the third year are the other 80% seeds dead, previously germinated or still dormant?
Line 48. What is meant by “Populations of this species can vary greatly…” What is varying?
Line 46 and 53 seem to contradict. Most seeds are dormant (line 46) but seeds germinate when they come in contact with moist soil (line 53).
Line 80-81 Why does actively being fed by carbonate groundwater make calcareous fens one of the most vulnerable ecosystems to climate change? Explain.
Line 75 – 95. This paragraph contains a mixture of different topics. Each main topic should be separated into its own paragraph. Then the main point being made can be clearly explained.
Line 100. I don’t know why this information is relevant to the study.
Line 107. Again, I don’t know why the information in this paragraph is relevant to the study.
Line 115 seems to indicate that the community is quite stable whereas back in line 48 it was noted that the populations of S. ferrugineus vary greatly and are not stable?
Line 127/128 note that the aim is to model the effects of climate change on S. ferrugineus but no introduction of the approach to be taken is included.
Comments on the Quality of English Language
While sentences were generally well written, the paper would be improved if the structure was clearer. I found the paragraph topics and opening sentences to be vague. If opening sentences focused on the general topic of the paragraph it would make the paragraph topics clearer.
Clearer paragraphs that made stronger connections between the approach and the aim would have aided my understanding of the approach used in the study. A clearer paragraph structure that emphasized the links between the methodology, results, and conclusions would have assisted.
Author Response
The basis for the study needs to be much more clearly explained in the introduction. The importance of the study is suggested, but there is no background on the general investigative approach to be used. This would be aided if paragraph topics were clearer and opening sentences focused on the general topic of the paragraph.
Answer: Dear reviewer. Thank you for your careful review of our article! Your recommendations and comments are very useful and improved the quality of manuscript. We have significantly revised the Introduction section, described the background on the general investigative and the choice of method in more detail (corrections are highlighted in green in the text of the article).
Insufficient explanation is given to make the methods section interpretable. For example: MaxEntv3.4.4 was used to assess the change of habitat conditions under climatic changes (Line 374). How was it used? What were the input variables and data sources? Explain the rationale for using it. Making the structure of the approach clearer would then make the results easier to follow. It would then be easier to see why the conclusions are justified.
Answer: Thanks for comment. We have relocated the Materials and Methods section, putting it before the Results. This made it easier and more consistent to present the data. We also described in the Introduction section the advantages of the maximum entropy method and made the Materials and Methods section more informative.
Other comments from the introduction:
Line 32 Rather than “They are characterized…” be specific about what is being referred to, i.e. These These calcareous mires are characterized…
Answer: Thank you! Done.
Line 46. What is the fruiting rate a proportion of? Do 10 – 32% of flowers form a fruit? If 20% of seeds germinated in the third year are the other 80% seeds dead, previously germinated or still dormant?
Answer: Fully formed seeds are found in only 10-32 % of flowers. The remaining seeds do not reach maturity. The most seeds are dormant in spikelets for more than one year, and 20% of seeds may germinate even in the third year. We have added the necessary explanations to the text.
Line 48. What is meant by “Populations of this species can vary greatly…” What is varying?
Answer: Thank you for the comment! Populations size of this species can vary greatly depending on weather conditions. We have added the necessary explanations to the text.
Line 46 and 53 seem to contradict. Most seeds are dormant (line 46) but seeds germinate when they come in contact with moist soil (line 53).PRIMARY COMMENTS:
Answer: Most seeds are saved for the second and third year as they remain in the spikelets.
Line 80-81 Why does actively being fed by carbonate groundwater make calcareous fens one of the most vulnerable ecosystems to climate change? Explain.
Answer: Thank you for the comment! The calcareous fens are actively fed by carbonate groundwater, and are one of the most vulnerable ecosystems to climate change. With increasing summer drought conditions, groundwater levels may decline, peat salinity may increase and the wetland complex will degrade with the establishment of new plant communities. We have added the necessary explanations to the text.
Line 75 – 95. This paragraph contains a mixture of different topics. Each main topic should be separated into its own paragraph. Then the main point being made can be clearly explained.
Answer: We completely agree with your comment. This paragraph has been revised.
Line 100. I don’t know why this information is relevant to the study.
Answer: Thank you for the comment. We have removed the paragraph describing secondary habitats of S. ferrugineus.
Line 107. Again, I don’t know why the information in this paragraph is relevant to the study.
Answer: Since the potential range has been analyzed, it necessary to indicate in which plant communities S. ferrugineus occurs at the eastern limit of its range. These data make it possible to use the floristic classification to compare communities involving S. ferrugineus in different parts of this species range range.
Line 115 seems to indicate that the community is quite stable whereas back in line 48 it was noted that the populations of S. ferrugineus vary greatly and are not stable?
Answer: The floristic composition of these communities is similar to that at the northern limit of the S. ferrugineus distribution, which is caused by carbonate-rich groundwater, which is caused by carbonate-rich groundwater. However, the abundance of S. ferrugineus local populations can vary greatly depending on the weather conditions of the current year.
Line 127/128 note that the aim is to model the effects of climate change on S. ferrugineus but no introduction of the approach to be taken is included.
Answer: We have significantly revised the Introduction section where we describe in more detail the benefits of the approach to be used.
Comments on the Quality of English Language
While sentences were generally well written, the paper would be improved if the structure was clearer. I found the paragraph topics and opening sentences to be vague. If opening sentences focused on the general topic of the paragraph it would make the paragraph topics clearer.
Clearer paragraphs that made stronger connections between the approach and the aim would have aided my understanding of the approach used in the study. A clearer paragraph structure that emphasized the links between the methodology, results, and conclusions would have assisted.
Answer: Thank you again for your favorable review and valuable recommendations. We have significantly revised the Introduction, Materials and Methods and Conclusions sections. We corrected the English translation of the manuscript, but if the quality of the translation is still not good enough, we agree to use MDPI's editing services.

Reviewer 2 Report
Comments and Suggestions for Authors
The work is very interesting, in content and approach. However, there are some remarks I have made in the margins of the article (written in red), which could improve the work.
There are also some small errors in the figures and text (highlighted).
Although I am not a native English speaker, the English is fluent and understandable.

Author Response
Comments and Suggestions for Authors
The work is very interesting, in content and approach. However, there are some remarks I have made in the margins of the article (written in red), which could improve the work.
There are also some small errors in the figures and text (highlighted).
Although I am not a native English speaker, the English is fluent and understandable.
Dear Reviewer!
Thank you for careful review and valuable comments! Your recommendations are very useful and improved the quality of our manuscript.
All necessary corrections have been made and marked in blue color in the text of the manuscript.
We have restated the aim of the study more clearly and described the perspectives for further research in more detail in the Conclusion section. We have also relocated the Materials and Methods section, putting it before the Results section.
